# Dolomites as SO$_2$ Sorbents in Fluid Combustion Technology

**Elżbieta Hycnar [1],\*, Tadeusz Ratajczak [2] and Magdalena Sęk [1]**

[1]    Department of Mineralogy, Petrography and Geochemistry, University of Science and Technology in Cracow, 30-059 Cracow, Poland; mmsek@agh.edu.pl

[2]    Mineral and Energy Economy Research Institute of Polish Academy of Science in Cracow, 31-261 Cracow, Poland; tarat@min-pan.krakow.pl

\*    Correspondence: hycnar@agh.edu.pl; Tel.: +48-12-617-4142

**Abstract:** Dolomites are not used as SO$_2$ sorbents in fluid combustion technology. The literature data show fundamental discrepancies in the possibility of such use. They mainly concern the role of magnesium in the sorption process of SO$_2$ and the durability of desulfurization products under high-temperature conditions. The article demonstrates that MgO is actively involved in the SO$_2$ binding under fluidized furnace conditions. The resulting products of sulfation contain magnesium in their compositions, and their thermal transformations begin only after the temperature exceeds 1100 °C. It has been shown that dolomites are a potential raw material for the production of SO$_2$ sorbents for fluid combustion technology, and their use is justified due to the higher desulfurization efficiency. Parameters of dolomite descriptions were given, by which it will be possible to predict the effects of flue gas desulfurization before the dolomites' use in industrial conditions. It has been shown that there are opportunities to expand the domestic raw-material base for the production of SO$_2$ sorbents, based on both dolomite resources present in deposits and dolomite waste accumulated in dumps, as well as generated during the current exploitation and processing of dolomites.

**Keywords:** dolomite sorbent; desulfurization; role of Mg in absorption SO$_2$; texture parameters and the efficiency of SO$_2$ absorption

## 1. Introduction

Poland is a country rich in dolomites. The national geological balance resources of dolomites as at 31 December 2019 amounted to 498.9 million tonnes, of which 204.8 million tonnes are the resources of exploited deposits (which constitutes 41.0% of the balance resources). The prospective resources are estimated at 504.2 million tonnes. The most economically important are: Precambrian marble dolomites occurring in the Sudety; Middle Devonian and Triassic dolomites in the Silesian–Cracow region; and dolomites of the Middle Devonian in the Świętokrzyskie Mountains. The forecast resources are estimated at 504.2 million tons. The mentioned occurrences form a rock series extending over large areas and characterized by a variety of mineralogical and chemical characters. This is reflected in the variability of quality parameters within the documented deposits, imposing the need to separate areas suitable for different applications [1,2].

Dolomites are a typical multi-raw material with various possibilities of use. In the national "balance of mineral resources" [3], deposits of dolomite, depending on the documented use, are separated into industrial (applicable in metallurgy, the glass industry, i.e., the dolomite meal), ceramic, and agricultural (as fertilizer dolomite and dolomite feed) uses, and also for the production of calcined dolomite (used in the fireproof industry), and crushed and block stones (used in construction and road construction as building stone and crushed aggregate).

The highest quality parameters characterize dolomites, which are useful for the production of dolomite flour for the glass industry and for use as a flux in metallurgy. The main criteria for economic capability of the Regulation of the Minister of the Environment on mineral deposits (Dz. U. of 2001., No. 153, item. 1774, as amended) used for industrial dolomite deposits, outside the minimum deposits stand volume, are the ratio of the thickness of the overburden; the thickness of the bed; and the minimum content of MgO (min. 16% wt.), as well as $Fe_2O_3$, $Al_2O_3$, and $SiO_2$, which indicates suitability for a specific direction of industrial application. Twelve deposits of industrial dolomites have been documented. These deposits are mainly found in the Silesian–Cracow region (11 deposits for the metallurgical industry and fireproof materials) and one in Lower Silesia (dolomite marble in the Rędziny deposit for the glass industry). The numerous deposits with proven or potential usefulness of the mineral as an industrial dolomite are documented as dolomite or dolomite marble for the road and construction industries. These include several deposits of dolomite and dolomite–calcite marble in the Kłodzko region and several deposits of Triassic or Devonian dolomites in the Silesian–Cracow region [3]. It is appropriate to perform testing in order to determine their suitability as $SO_2$ sorbents used in power engineering.

Among the methods of heat and electricity production, fluid combustion technology (FBC) deserves special attention, as it allows the use of not only traditional fuels, but also all kinds of energy-carrying waste. It also guarantees a reduction in NOx and $SO_2$ emissions concurrently with fuel combustion. The modern fluidized furnaces give the possibility of producing water vapor not only with subcritical parameters, but also supercritical, high efficiency of the combustion process through the use of circulating (CFB) and pressure (PFBC, PCFB) fluid combustion technology, both with circulating and with stationary fluid layers. The main problem of the presented combustion technology is the desulfurization efficiency at the combustion stage in the furnace. In this case, the function of $SO_2$ absorbents (the new solid sulfate phases are formed) is played by high-quality limestones with a $CaCO_3$ content of at least 85% wt. Depending on the conditions in the boiler and the excess of sorbent used, the efficiency of $SO_2$ reduction is 30–60%. Higher efficiency, at the level of 80–90%, can be obtained when calcium hydroxide is used in the form of a sorbent. The use of $Ca(OH)_2$ in this case should be considered unjustified, mainly due to the high costs.

The literature data mention the possibility of using dolomites as $SO_2$ sorbents [4–13]. The authors of the study present three positions on the use of dolomites as $SO_2$ sorbents in fluid combustion technologies, in both atmospheric and pressure boilers:

1. The magnesium component (MgO) is treated as a non-reactive ballast [4,6,9,11]. This leads to their low desulfurization efficiency compared to limestones [13].
2. The share of magnesium in the $SO_2$ binding reaction is shown. However, the thermodynamic stability of desulfurization products containing magnesium in the structure is questioned under temperature conditions typical for fluidized bed furnaces, thus indicating lower sorption efficiency of dolomites [10,12,14].
3. The effective share of MgO in $SO_2$ binding is confirmed on the basis of the presence of both calcium and magnesium ($CaMg_2(SO_4)_3$), and magnesium sulfates ($MgSO_4$) among the desulfurization products, which indirectly indicates their durability under the experimental temperature conditions (up to 850 °C). The desulfurization efficiency of the dolomites demonstrated in this case is comparable to and even higher than that of the limestone [5,7,8,15].

The dolomites are considered as potential $SO_2$ sorbents in any case, but the results of the studies show significant discrepancies in both the role of magnesium in the $SO_2$ binding process and the thermodynamic durability of desulfurization products containing magnesium in their compositions. The research methodology of experimental work concerning the process of sulfation is also diversified. For these purposes, thermogravimetric analysis is commonly used [5,7]. In some cases, specially constructed experimental equipment is used, which approximates the conditions in fluidized furnaces [14]. There are different variants of this type of installation, and the sulfate process

itself is carried out under different conditions. For example, the gas composition (in terms of $SO_2$ or $CO_2$ concentration) used in the sulfation process is varied. The authors, comparing the $SO_2$ capture capacity of limestones and dolomites, as a rule ignore the influence of petrographic properties on the sulfur capture capacity [9].

The authors also note that under the conditions of fluidized furnaces, dolomite attrition was extensive compared to limestones [9,16]. They test the attrition of sorbents during the calcination process. The calcination and sulfation processes take place almost simultaneously in real conditions, which can significantly reduce the intensity of the surface abrasion and limit fragmentation of the sorbent grains. They also do not investigate the structural and textural nature of rocks and decarbonation products, which may play a decisive role in this case [17,18].

The aim of the research presented in the paper was to determine the possibility of using dolomites in the form of $SO_2$ sorbents in fluid combustion technology. During the research, particular attention was paid to the participation of magnesium in the $SO_2$ binding reaction and the durability of magnesium desulfurization products under high-temperature conditions. The parameters of these rocks responsible for the effectiveness of $SO_2$ sorption were also studied. The results of the research became the basis for an attempt to expand the domestic resource base with different types of minerals useful for the production of $SO_2$ sorbents for the needs of fluid combustion technology. As the dolomites exploited in the country represent age-diverse varieties (Precambrian, Paleozoic, Triassic, Silurian) that are also genetically varied (primary dolomites, secondary–resulting from the metasomatosis of limestone, dolomitic marbles, calcium dolomites, dolomitic limestone) thus having a varied mineral composition and structural–textural character, a research methodology was proposed that will allow us to predict the behavior of the sorbent in industrial conditions at the stage of laboratory tests.

## 2. Materials and Methods

The study used dolomites mined from deposits located in Poland: Lower Silesia–Romanowo Górne, Rędziny, Lesser Poland–Żelatowa, Upper Silesia–Chruszczobród II (Table 1, Figure 1). In the case of dolomites from the Romanowo Dolne deposit, the product of the fine regrind was the subject of the study. In other cases, rocks material derived from trials of operating walls or drilling cores were tested. For comparative studies we also covered industrial sorbent, made of limestone.

**Table 1.** The list of dolomites used for the evaluation of sorption properties in relation to $SO_2$ in the conditions of fluid combustion technology.

| Sample Number | 1. | 2. | 3. | 4. |
|---|---|---|---|---|
| Origin | Dolomite mined from the Rędziny deposit | Dolomite mined from the Romanowo Górne deposit | Dolomite mined from the Żelatowa deposit | Dolomite mined from the Chruszczobród II deposit |
| Age | Sylurian | Precambrian | Triassic | Triassic |

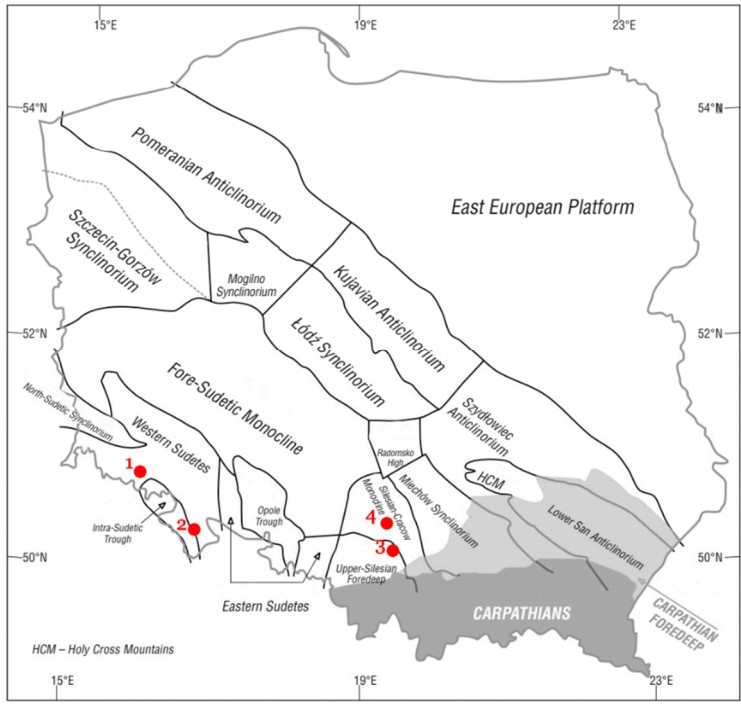

**Figure 1.** The locations of dolomite deposits from which materials were collected for research on the background the main tectonic units on the sub-Cenozoic surface of the Polish [19]. Explanations: 1—Rędziny deposit; 2—Romanowo Górne deposit; 3—Żelatowa deposit; 4—Chruszczobród II deposit.

The sulfation experiment was carried out based on the guidelines developed by Ahlstrom Development Laboratory [20]. This method is based on determining two indicators: the reactivity (RI index) and absolute sorption (CI—capacity index). The reactivity index determines the ratio of the calcium content (in the case of dolomite studies, magnesium was additionally included) in the sample to the amount of sulfur after the sorption process (Ca/S moles). The absolute sorption index CI, in turn, determines the amount of sulfur sorbed by 1000 g of the sorbent (g S/1000 g of the sorbent). The $SO_2$ sorption studies were carried out using a material with a particle size of 0.125–0.250 mm. The sulfation experiment was carried out on the basis of an instance designed on the basis of a gas-tight retort furnace acting as a fixed bed. As required, the samples were subjected to a decarbonation process at 850 °C for 30 min prior to the sulfation. A sample of the sorbent mass of 150 mg was placed inside the combustion chamber, on a perforated ceramic plate, in such a way that the individual grains of the sorbent were not in contact with each other. In this way, free access of gases to individual sorbent grains was ensured during the experiment. The first, synthetic, air containing 80% $N_2$ and 20% $O_2$ was passed through it. Then, a gas containing 1780 ppm of $SO_2$, 3% of $O_2$, 16% of $CO_2$, and $N_2$ making up the rest was passed through the samples with a speed of 950 mL/s for another 30 min. In the next stage, the content of absorbed sulfur was determined using an elemental analysis apparatus for carbon, hydrogen, nitrogen, and sulfur (Series 628) by LECO. The results of the research became the basis for calculating the values of reactivity (RI) and absolute sorption (CI) indicators according to the formula:

$$RI = \frac{\frac{x_{Ca}}{100} \cdot \frac{M_S}{M_{Ca}} \cdot \left(1 - \frac{M_{CO_2}}{M_C} \cdot \frac{x_{C_P}}{100} - \frac{M_{SO_3}}{M_S} \cdot \frac{x_{S_P}}{100}\right)}{\frac{x_{S_P} - x_{S_b}}{100} + \frac{M_{CO_2}}{M_C} \cdot \left(\frac{x_{C_P} \cdot x_{S_b} - x_{C_b} \cdot x_{S_P}}{10000}\right)} \tag{1}$$

$$CI = \frac{1000 \cdot \left[\frac{x_{S_P} - x_{S_b}}{100} + \frac{M_{CO_2}}{M_C} \cdot \left(\frac{x_{C_P} \cdot x_{S_b} - x_{C_b} \cdot x_{S_P}}{10000}\right)\right]}{1 - \frac{M_{CO_2}}{M_C} \cdot \frac{x_{C_P}}{100} - \frac{M_{SO_3}}{M_S} \cdot \frac{x_{S_P}}{100}} \tag{2}$$

Explanations: $x_{Ca}$, $x_{C_p}$, $x_{S_p}$, $x_{C_b}$, $x_{S_b}$—percentages of calcium in the sorbent, carbon in the sorbent after the sulfating process, sulfur after the sulfating process, carbon before the sulfating process, and sulfur before the sulfating process, respectively (%); $M_S$, $M_{Ca}$, $M_C$, $M_{CO_2}$, $M_{SO_3}$—molar masses of sulfur, calcium, carbon, carbon dioxide, and sulfur trioxide, respectively (kg/kmol).

To evaluate the sorption capacity, the five-level scale proposed by Ahlstrom Development Laboratory was used (Table 2).

**Table 2.** The reference values of the reactivity (RI) (Ca moles/S moles) and the absolute sorption (CI) (g S/1000 g of the sorbent) [20].

| The Sorption Capacity of the Sorbent | RI (kmol Ca/kmol S) | CI (g S/1 kg of Sorbent) |
|:---:|:---:|:---:|
| Excellent | <2.5 | >120 |
| Very good | 2.5–3.0 | 100–120 |
| Good | 3.0–4.0 | 80–00 |
| Sufficient | 4.0–5.0 | 60–80 |
| Low quality | >5.0 | <60 |

In the next stage of the study, an attempt to define the sorbent's parameters affecting its reactivity was made. The analyses were aimed at:

- Determining the mineral composition, petrographic nature, and structural and textural characteristics of the examined limestones. For this purpose, X-ray diffraction (MiniFlex 600, Rigaku, Tokyo, Japan), optical (BX51, Olympus, Tokyo, Japan) and scanning microscopy (Quanta 200 FEG, FEI, Hillsboro, OR, USA) were applied.

- Chemical composition. Apart from the quantitative determination of $CaCO_3$ and $MgCO_3$, the content of $SiO_2$, $Al_2O_3$, $K_2O$, $Na_2O$, $Fe_2O_3$, $Mn_2O_3$, and $TiO_2$ was also tested. The research was performed with the use of atomic emission spectroscopy (ICP–OES Plasma 40, Perkin–Elmer, Waltham, MA, USA). The samples were mineralized using a high-pressure microwave mineralizer (Speedwave Xpert, Berghof, Eningen, Germany). Mineralization was carried out in a closed system.

- Temperature and decomposition degree and the thermal dissociation course of dolomites process (STA 449 F3 Jupiter + QMS 403C Aelos, Netzsch, Selb, Germany). The research was carried out using thermogravimetry (TG) and differential thermal analysis (DTA). The samples were analyzed in the temperature range of 20–1000 °C. The heating rate was 20 °C/min. The measurements were made in the air atmosphere.

- A porous texture analysis was performed using mercury porosimetry. The following porous texture parameters were determined:

  1. The coefficient of effective porosity, that is the ratio of pore volume to the total volume of the sample [21–24]:

  $$\phi = \frac{V_{tot}}{V_b} \cdot 100\% = \frac{V_b - V_s}{V_b} \cdot 100\% = \left(1 - \frac{\rho_b}{\rho_s}\right) \cdot 100\% \tag{3}$$

  Explanations: $\phi$—the coefficient of effective porosity (%); $V_{tot}$—the total volume of mercury in the pores (mL); $V_b$—external volume (mL); $V_s$—skeleton volume (mL); $\rho_b$—bulk density (g/mL); $\rho_s$—skeletal density (g/mL).

  2. The specific surface area of porous space, that is, the pore area in relation to the sample unit mass. This parameter characterizes the flow resistance of reservoir media in the porous medium. The specific surface area, assuming the reversibility of the injection process is determined on the basis of the obtained pore volume according to the following equation [21,24]:

  $$A = -\frac{1}{\gamma \cos \theta} \int_0^V P dV \tag{4}$$

Explanations: A—the total surface area of porous space ($m^2$/g); dV—partial pore volume corresponding to the given capillary pressure ($m^3$); P—capillary pressure (psi); $\gamma$—surface tension of mercury (dyna/cm); $\theta$—contact angle (°).

3. The average pore diameter ($D_{av}$) is expressed with the weight of the pore size for the entire pore diameter range in the sample and is calculated using the following equation [21,23]:

$$D_{av} = \frac{4 \cdot V_{tot}}{A} \tag{5}$$

Explanations: $D_{av}$—the average pore diameter ($\mu$m); $V_{tot}$—the total volume of mercury in the pores (mL); A—total specific surface area of the pore space ($m^2$/g).

In order to characterize the decarbonation and $SO_2$ sorption processes, the mentioned texture parameters were determined for both the decarbonizated and sulfated samples.

- Phase composition and textural nature of the resulting desulfurization products. X-ray diffractometer (Rigaku MiniFlex 600) and a scanning microscope (FEI Quanta 200 FEG) were used for the tests.

An attempt was made to demonstrate the stability of desulfurization products under high-temperature conditions based on DTA and TG (TGA / DSC 3+, Mettler Toledo, Greifensee, Switzerland) in the temperature range up to 1200 °C. For this purpose, the characteristic melting temperatures of desulfurization products in both oxidizing (air) and reducing atmosphere (mixture of CO and $CO_2$ in a volume ratio of 3:2) were also determined based on PN-G-04535: 1982 [25]. The observations were conducted by heating the test material to a temperature of 1500 °C.

The research was carried out with the use of research equipment of the AGH University of Science and Technology in Cracow with the support of the Energy Center AGH.

## 3. Results and Discussion

### 3.1. Phase and Chemical Composition of the Dolomites Tested

The material used in the studies represents high-quality dolomites with a $CaCO_3$ and $MgCO_3$ at the level of 48–57% wt. and 40–49% wt. respectively (Table 3). The dolomite is by far the dominant mineral component in relation to both calcite and other minerals.

Dolomites from the Rędziny deposit represent dolomitic marbles with a characteristic granoblastic structure and a clearly marked parallel texture. The main component of the rock is dolomite, which forms crystals with a clearly elongated structure, showing the typical multiple-twins of this mineral (Figure 2a). In addition, scattered in the rock are a few, small enclaves filled with fibrous serpentine mineral, probably chrysotile (Figure 2b). Additionally, a few talcum crystals of a lamellar shape were identified between the dolomite crystals. The total share of non-carbonate minerals was about 5% by volume of rocks [26].

**Table 3.** Chemical composition of the tested dolomites and industrial sorbent (% wt.).

| Component | Dolomite from the Rędziny Deposit (1) | Dolomite from the Żelatowa Deposit (2) | Dolomite from the Romanowo Górne Deposit (3) | Dolomite from the Chruszczobród II Deposit (4) | Industrial Sorbent (5) |
|---|---|---|---|---|---|
| $SiO_2$ | 0.15 | 0.32 | 2.87 | 0.82 | 0.53 |
| $TiO_2$ | >0.001 | 0.050 | >0.001 | 0.006 | >0.001 |
| $Al_2O_3$ | 0.25 | 0.22 | 0.13 | 0.06 | 0.28 |
| $Fe_2O_3$ | 0.48 | 0.86 | 0.09 | 0.53 | 0.24 |
| CaO | 27.54 | 32.01 | 31.57 | 26.94 | 54.58 |
| MgO | 23.56 | 19.28 | 19.12 | 23.06 | 0.33 |
| MnO | 0.07 | 0.08 | 0.04 | 0.023 | 0.011 |
| $Na_2O$ | 0.04 | 0.06 | 0.14 | 0.035 | 0.014 |
| $K_2O$ | 0.007 | 0.01 | 0.08 | 0.006 | 0.005 |
| $P_2O_5$ | 0.01 | 0.02 | 0.01 | 0.055 | 0.028 |
| Ignition loss | 47.87 | 47.07 | 45.59 | 48.16 | 43.78 |
| SUM | 99.99 | 99.98 | 99.64 | 99.59 | 99.80 |
| $CaCO_3$ | 49.02 | 56.99 | 56.19 | 48.06 | 97.37 |
| $MgCO_3$ | 49.00 | 40.10 | 39.78 | 47.97 | 0.68 |
| Sum of carbonate | 98.03 | 97.08 | 95.96 | 96.03 | 98.05 |

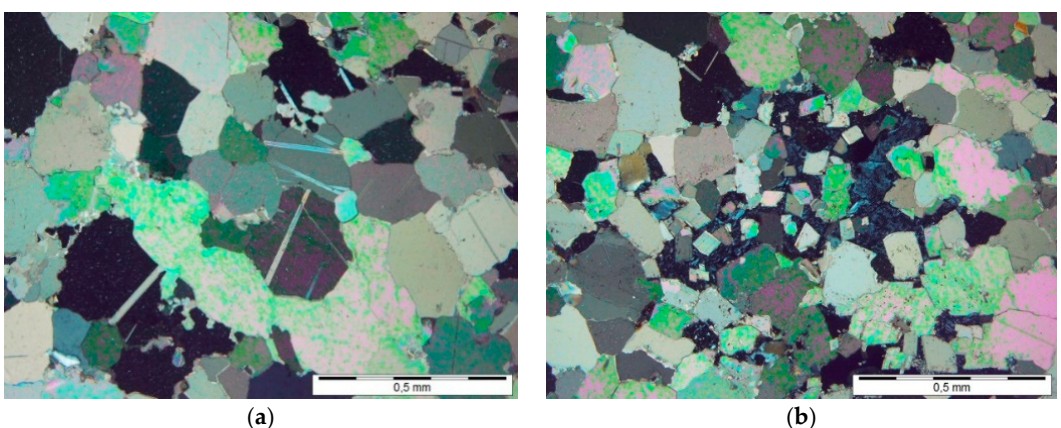

(**a**)          (**b**)

**Figure 2.** The dolomites from the Rędziny deposit: (**a**) anhedral, often repeatedly twinned crystals dolomite; (**b**) irregular enclave filled with serpentine minerals (grey colours). Polarizing microscope, Xp.

In the mineral composition of dolomites from the Romanowo Upper deposit, in addition to the dolomite, the presence of calcite and small amounts of quartz, albite, vermiculite, and flogopite were found (Figure 3). The mineral composition and, above all, the presence of vermiculite and flogopite indicates that these dolomites are also the result of poorly advanced metamorphic transformations.

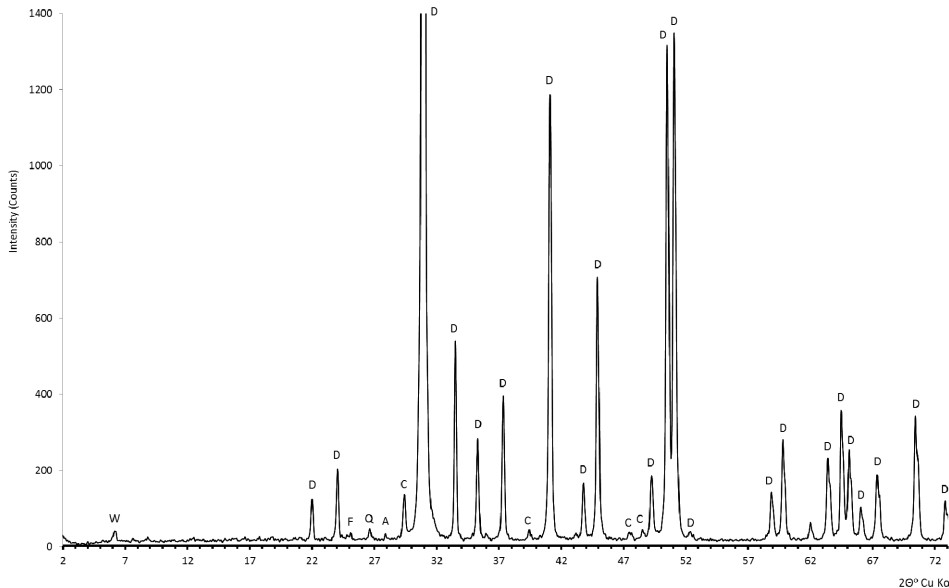

**Figure 3.** The phase composition of dolomite from the Romanowo Górne deposit is shown in the diffraction pattern. Explanations: A—albite ($NaAlSi_3O_8$), D—dolomite ($CaMg(CO_3)_2$); C—Calcite ($CaCO_3$); F—flogopite ($K_2(Mg,Fe^{2+})_6[(OH,F)_4/Al_2Si_6O_{20}]$); W—vermiculite ($(Mg,Ca)(Mg,Fe^{3+},Al)_6[(OH)_4/Al,Si)_8O_{20}]\cdot 8H_2O$); Q—quartz ($SiO_2$).

In the dolomites from the Chruszczobród II deposit, the dolomite that builds the rock forms two generations of crystals. The first of these are microcrystalline crystals, occurring in spherical shapes clusters, corresponding to the relics of ooid or oncolytic forms, characteristic of limestone (Figure 4a). The second generation is crystals that are clearly larger, often with the likes of rhomboedric (Figure 5b), showing zonal and skeletal structure. It should be assumed that these are secondary (metasomatic) dolomites resulting from the limestone dolomitization process. The process of calcite dolomitization took place under the influence of magnesium-rich waters circulating through crevices, which caused dissolution of calcium carbonate, partial discharge of $Ca^{2+}$ ions, and their replacement with $Mg^{2+}$ ions. In the final stage of the dolomitization process, the water was additionally rich in iron, which resulted in the formation of iron-enriched dolomite—$Ca(Mg,Fe)(CO_3)_2$ (Figure 6).

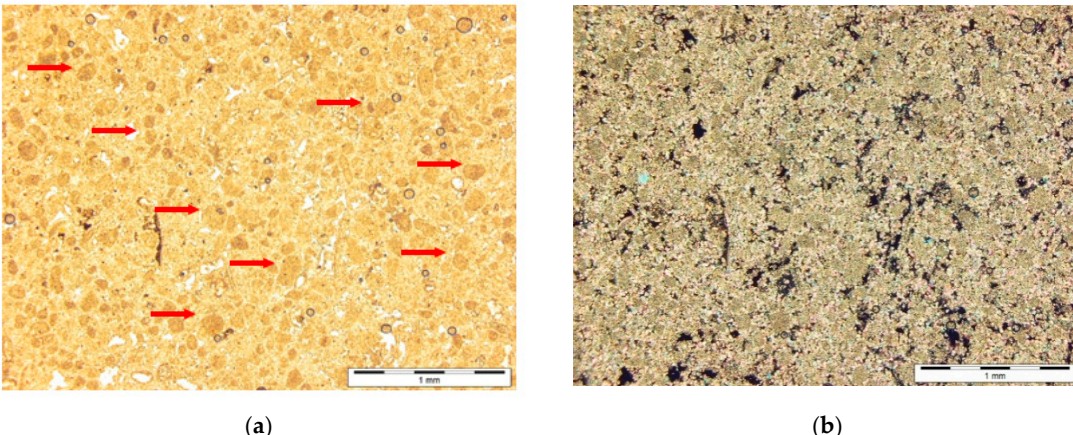

(**a**)   (**b**)

**Figure 4.** The dolomites from the Chruszczobród II deposit: (**a**) the relics of allochemic components forming the skeleton of the rock (arrow); (**b**) the rock pores' opal filling. Polarizing microscope, Ip (**a**), Xp (**b**).

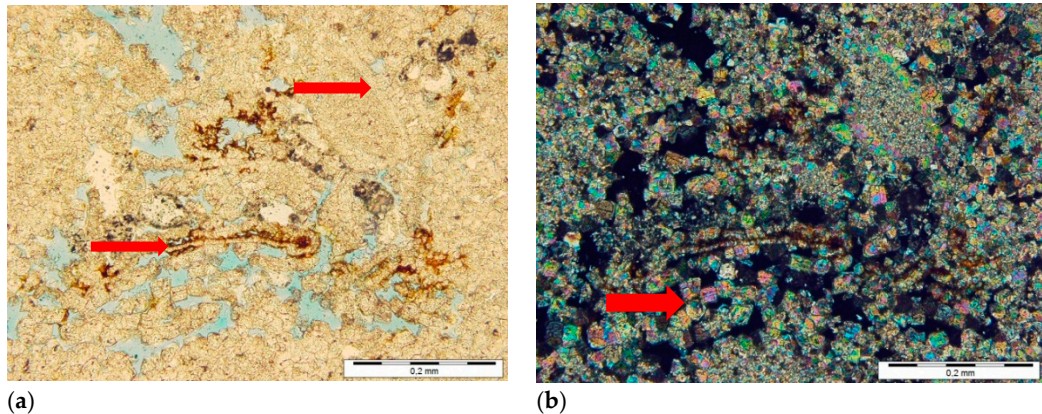

(**a**)        (**b**)

**Figure 5.** The dolomites from the Chruszczobród II deposit: (**a**) the relics of carbonate bioclasts (arrows), the iron compounds of the iron oxides–hydroxides nature (brown colour), rock pores stained with blue dye; (**b**) the dolomite crystals of a rhombohedral habit (arrow). Polarizing microscope, Ip (**a**), Xp (**b**).

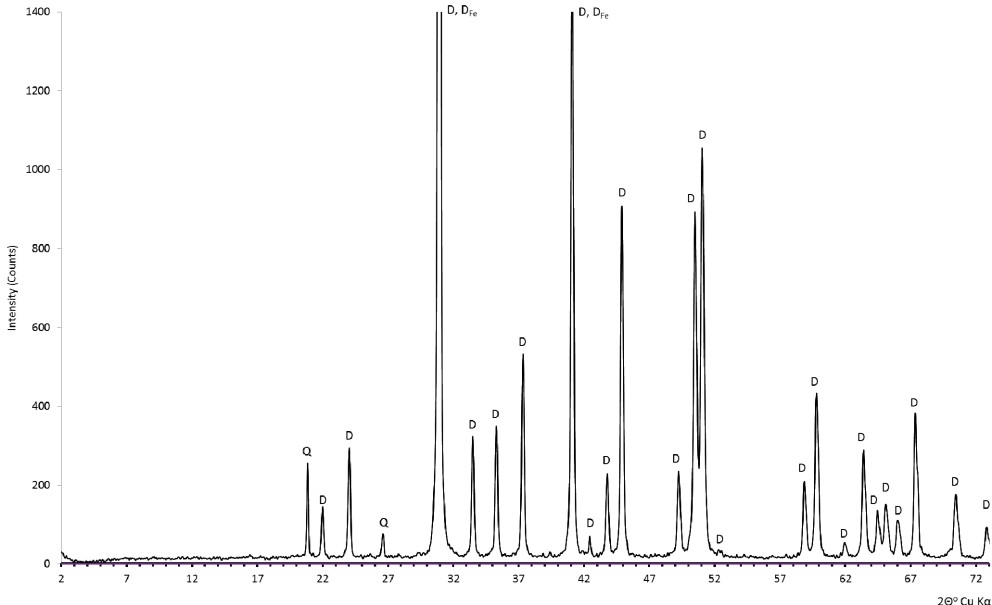

**Figure 6.** The phase composition of dolomite from the Żelatowa deposit is shown on the diffraction pattern. Explanations: D—dolomite ($CaMg(CO_3)_2$); $D_{Fe}$—ferruginous dolomite $Ca(Mg,Fe)(CO_3)_2$; Q—quartz ($SiO_2$).

In addition to the dolomite, opal and chalcedon were found in small quantities, filling rock pores mainly (Figure 4b), less commonly in the form of concretions or bioclasts—sponge needles (Figure 5a). The pyrite framboidal occurrences, bearing clear traces of oxidation and concentration of kryptocrystalline iron compounds, found in the pores and surfaces of dolomite crystals were also identified (Figure 5a). The described mineral phases indicate the presence of secondary mineralization processes, mainly silification and pyritization [27].

The calcite has been identified in the dolomites from the Żelatowa deposit, apart from the dolomite. The content of dolomite relative to the calcite is variable and these rocks can be classified as dolomites, calcareous dolomites, and even dolomitic limestones. Microscopic observations in cathodoluminescence (CL) revealed the presence of allochem components characteristic of limestones, such as ooids and pellets, which recrystallized during diagenesis. (Figure 7a,b). In their mineral composition, calcite occurs next to dolomite. Between them there is an abundant block cement,

made of dolomite. The calcite and dolomite cleavage fissures and crystal surfaces are covered iron hydroxide compounds, with a characteristic brown color [26]. Dolomites from the Żelatowa deposit, like the dolomite from the Chruszczobród II deposit, were also formed as a result of metasomatic transformations of limestones. The diversified advancement of the dolomitization process is the result of the coexistence of limestones and dolomites in the Żelatowa deposit, as well as transition links between these rocks—calcareous dolomites and dolomitic limestones.

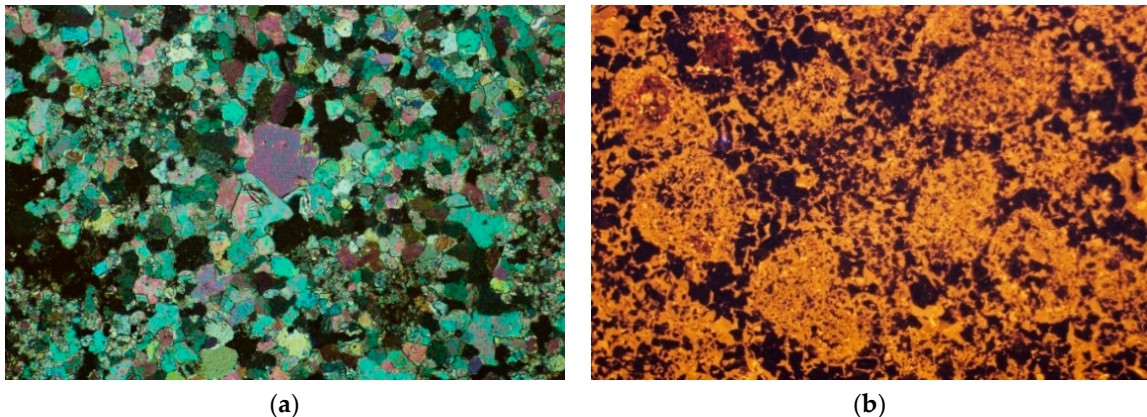

| (**a**) | (**b**) |

**Figure 7.** The dolomites from the Żelatowa deposit. The visible recrystallized carbonate grains; (**a**) polarizing microscope, Xp; (**b**) cathodoluminescence (CL).

### 3.2. The Sorption of Efficiency SO₂

The values of the CI and RI indexes, presented in Table 4, allowed us to assess the sorption properties of the studied dolomites as excellent in the scale presented in Table 2 (RI < 2.5; CI > 120). These values suggest that the dolomites can be treated as a potential raw material for the production of $SO_2$ used in fluid combustion technology. Comparing the RI and CI values of dolomites with industrial sorbent, which is the best product of this type on the domestic market, it can be concluded that sorbents produced from dolomites will be characterized by a higher $SO_2$ binding efficiency compared to the limestone milling products available on the market. The RI and CI values of the studied dolomites are much more favorable than the analyzed industrial sorbent.

**Table 4.** The values of the absolute sorption (CI) (g S/1 kg sorbent) and reactivity (RI) (kmol Ca/kmol S) indicators of the dolomites and industrial sorbent tested.

| Indicator | Dolomite from Rędziny Deposit (1) | Dolomite from Żelatowa Deposit (2) | Dolomite from Romanowo Górne Deposit (3) | Dolomite from Chruszczobród II Deposit (4) | Industrial Sorbent (5) |
|---|---|---|---|---|---|
| CI | 197 | 180 | 168 | 174 | 130 |
| RI | 1.52 | 2.05 | 2.10 | 1.80 | 2.35 |

In order to confirm the high efficiency of $SO_2$ binding by the studied dolomites, the degree of individual dolomite grains conversion was compared with the industrial sorbent. In the photographs, sorbent grains were switched in cross-section after the sulfate process (Figure 8a,b). The interiors of dolomite grains have been evenly sulfated (Figure 8a), in contrast to the industrial sorbent. In their case, only the outer part of the grains are characterized by a high degree of conversion; the inside of the sorbet doesn't react with $SO_2$ (Figure 8b).

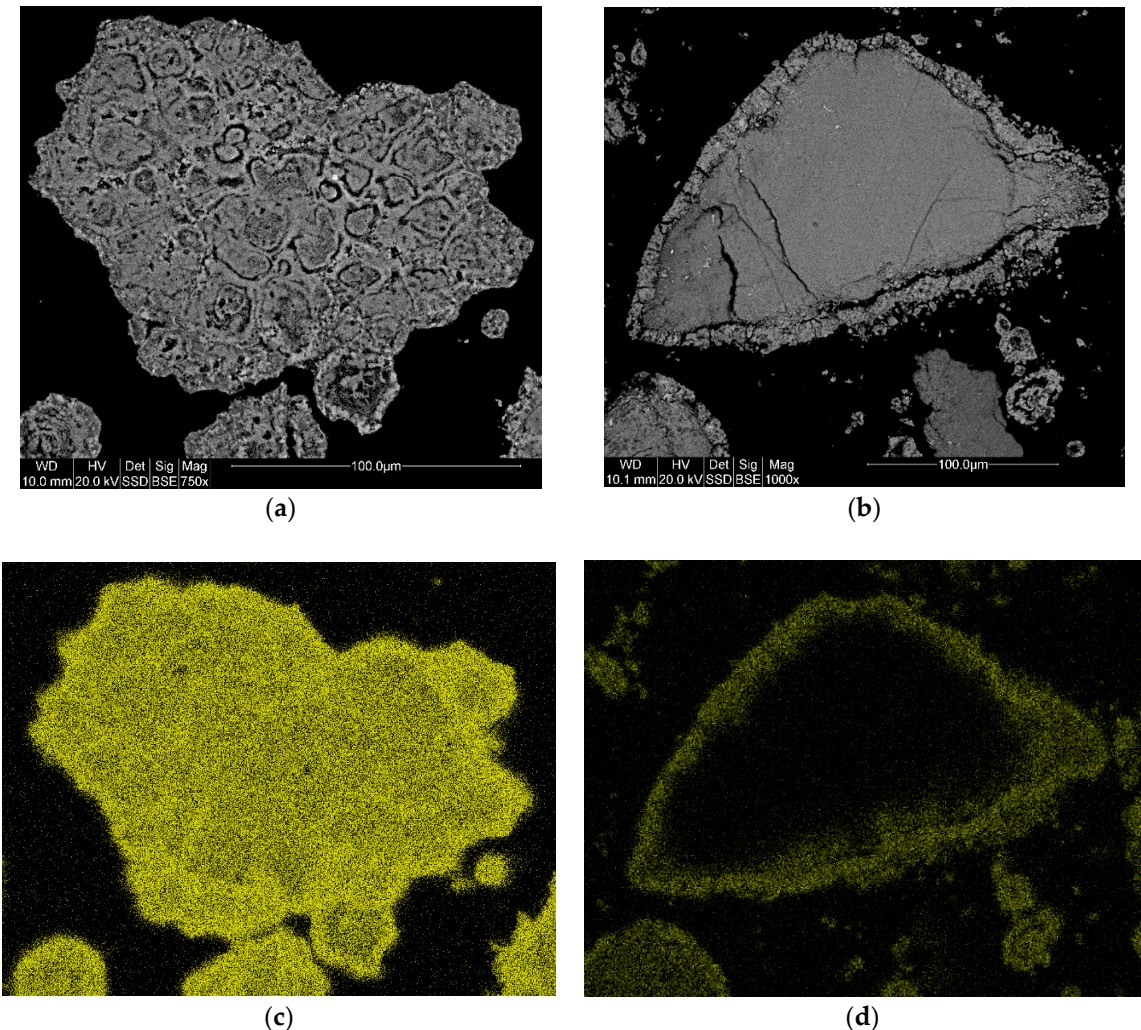

**Figure 8.** The SEM images of dolomite grain from the Chruszczobród II deposit (**a**,**c**) and industrial sorbent produced from limestones (**b**,**d**) in cross-section after the sulfation process in the light of backscattered electrons. Visible: (**a**,**c**)—evenly sulfated interior of the dolomite grain; (**b**,**d**)—non-sulfated grain interior of industrial sorbent; (**c**,**d**)—the S distribution is marked in yellow (SEM/EDX).

*3.3. The Dolomite Parameters Responsible for the Efficiency of $SO_2$ Sorption*

The influence on the efficiency of $SO_2$ sorption in fluidized bed boilers, in addition to the chemical and phase composition having:

- The course and temperature of calcite/dolomite thermal dissociation;
- the structural and textural nature of the sorbent, and above all its porosity, which shapes the course of the sulfation process, both in laboratory and industrial conditions [28].

The dolomite decarbonation process, regardless of the origin of this mineral, is in two stages (Table 5, Figure 9). The first step involves the decomposition of $CaMg (CO_3)_2$ into CaO and MgO and the recombination of some gaseous $CO_2$ with CaO to form secondary $CaCO_3$ (calcite). In the second stage, the calcite formed is decomposed into CaO and $CO_2$.

**Table 5.** The thermal effects occurring during the heating of the dolomite in the temperature range up to 1000 °C on the example of the dolomite from the Chruszczobród II deposit.

| Type of Process | Type of Reaction | Process/Reaction Temperature | | Change of Weight |
|---|---|---|---|---|
| | | Beginning | End | |
| | | (°C) | (°C) | (% by wt.) |
| Evaporation of surface water and hygroscopic moisture | – | 25 | 600 | – |
| Decomposition of $CaMg(CO_3)_2$ | endothermic | 600 | 790 | 21 |
| Formation of secondary $CaCO_3$ | exothermic | 790 | 840 | 40 |
| Decomposition of $CaCO_3$ | endothermic | 840 | 890 | 47 |
| – | – | 890 | 1000 | - |

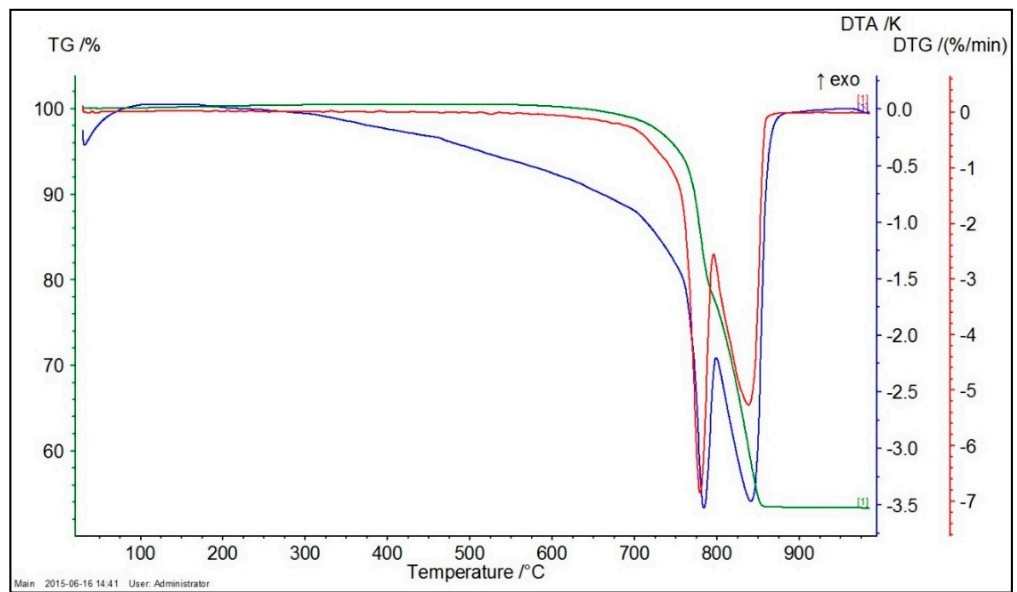

**Figure 9.** The thermal curves of differential thermal analysis (DTA), thermogravimetry (TG), and derivative thermogravimetry (DTG) of dolomite from the Chruszczobród II deposit, showing the course of the thermal dissociation process.

In the case of the studied dolomites, the decomposition of $CaMg(CO_3)_2$, related to the loss of $CO_2$ from the crystal structure, begins after exceeding the temperature of 600 °C. At the temperature of 850 °C, the degree of decarbonation is high and amounts to 39.51–45.58% wt. Increasing the temperature to 1000 °C doesn't cause significant changes in mass, and the loss at the level of 3.0–3.5% wt. is caused by the decomposition of calcite, both originally present in the mineral composition of the rock, and secondary, formed as a result of recombination of CaO and $CO_2$ resulting from the decomposition of the dolomite (Table 6, Figure 10).

**Table 6.** Weight loss during heating of the tested dolomites (% wt.).

| Temperature Range | Dolomite from Deposit | | | |
|---|---|---|---|---|
| | Rędziny (1) | Żelatowa (2) | Romanowo Górne (3) | Chruszczobród II (4) |
| 25–850 °C | 43.22 | 38.40 | 42.02 | 45.92 |
| 25–1000 °C | 44.40 | 41.84 | 44.51 | 47.02 |
| The heating 850 °C for 0.5 h | 44.81 | 39.51 | 43.12 | 45.58 |

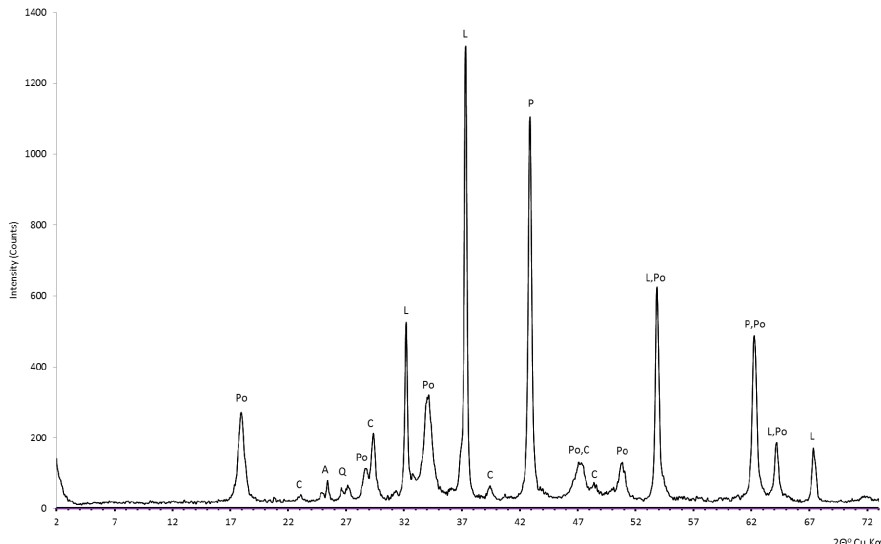

**Figure 10.** The phase composition of dolomites from the Chruszczobród II deposit after the decarbonation process at the temperature of 850 °C is shown in the diffraction pattern. Explanations: A—anhydrite ($CaSO_4$), C—calcite ($CaCO_3$), L—lime (CaO), P—periclase (MgO), Po—portlandite ($Ca(OH)_2$), Q—quartz ($SiO_2$).

The two-stage process of decarbonation has a decisive impact on the dolomite porosity development. Table 7 presents exemplary parameters of the porous texture of dolomites (based on the example of the dolomite from the Chruszczobród II deposit) in relation to the industrial sorbent produced from limestone, determined using the mercury porosimetry. Using this method, pores with diameters greater than 5 nm are measured, excluding the range of micropores and some of the mesopores.

Differences can also be observed in the average pore diameter between the dolomite and the industrial sorbent. The detailed information in this case is provided by the pore volume distributions as a function of the diameters presented in Figures 11 and 12. The secondary porosity of dolomite (on the example of dolomite from the Chruszczobród II deposit—Figure 12), important for the $SO_2$ binding process, is based on a wider range of pore diameters (from approx. 0.03–11 μm) compared to industrial sorbent, which develops porosity mainly based on pores with diameters in the narrow range of 0.3–6.0 μm (Figure 11).

**Table 7.** The parameters of the porous texture of dolomite from the Chruszczobród II deposit and industrial sorbent produced from limestone, determined with the use of mercury porosimetry.

| Sample Symbol | The Total Pore Volume $V_{POR}$ ($cm^3$) | The Average Pore Diameter $D_{POR}$ ($\mu m$) | The Specific Surface $S_{POR}$ ($m^2/g$) | The Effective Porosity $P_{POR}$ (%) | |
|---|---|---|---|---|---|
| Dolomite form the Chruszczobród II deposit | | | | | |
| 1 | 0.57 | 3.77 | 0.47 | 54.87 * | 9.61 ** |
| 2 | 1.97 | 0.61 | 18.57 | 82.32 * | 42.76 ** |
| 3 | 0.46 | 9.02 | 0.21 | 55.39 * | 2.98 ** |
| Industrial sorbent | | | | | |
| 1 | 0.34 | 4.98 | 0.24 | 43.02 * | 1.00 ** |
| 2 | 0.93 | 0.45 | 3.14 | 70.17 * | 28.16 ** |
| 3 | 0.35 | 0.56 | 0.52 | 32.80 * | 4.18 ** |

Explanations: * For the full range of pore diameters: 0.0005–1000 $\mu m$; ** For the range of pore diameters: 0.01–10 $\mu m$; 1—natural sample, 2—after the decarbonation process, 3—after the sulfation process.

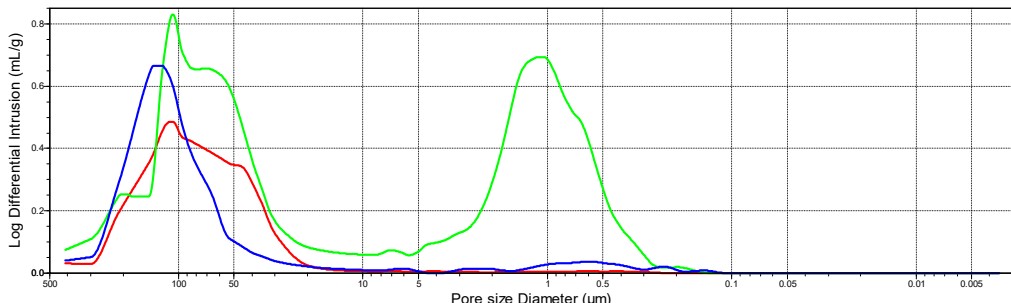

**Figure 11.** The distribution of pore volume as a function of the diameters of industrial sorbent made of limestone. Explanations: natural sample; sample after the thermal dissociation process; sample after the sulfation process.

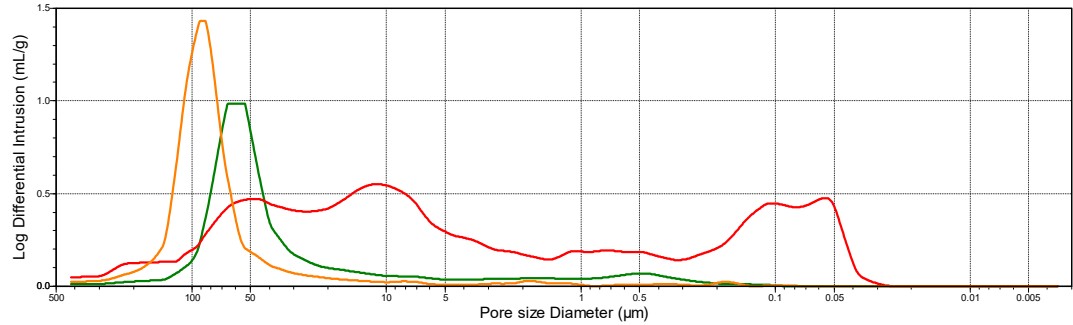

**Figure 12.** The distribution of pore volume as a function of the diameters of dolomite sorbent made from the Chruszczobród II deposit. Explanations: natural sample; sample after the thermal dissociation process; sample after the sulfation process.

The sorbent porosity, which is formed as a result of the release of $CO_2$ from the calcite/dolomite structure (the so-called secondary porosity), is directly responsible for the desulfurization efficiency, due to the fact that the $SO_2$ sorption process takes place on the internal surface of the pores formed during thermal dissociation [29]. Because the molar volume of the resulting desulfurization products in the form of $CaSO_4$, in the case of dolomite, also $CaMg_2(SO_4)_3$, is much greater with respect to $CaCO_3$ and $CaMg(CO_3)_2$, the reactivity of the sorbent will be determined by the specific surface area capable of reacting with $SO_2$. The surface capable of reacting with $SO_2$ should be considered one that has

been developed through pores of sufficiently large diameters, on the border of meso- and macropores (division according to International Union of Pure and Applied Chemistry).

In industrial conditions, where the processes of decarbonation and $SO_2$ sorption occur almost simultaneously, the primary porosity of the sorbent, which is an individual feature of the rock, shaped by processes occurring during sedimentation, diagenesis, and rock epigenesis, will also be important from the sorption properties point of view. These types of pores create diffusion channels of $CO_2$ from the inside and $SO_2$ to the inside of the sorbent grains, intensifying the processes of $SO_2$ decarbonation and sorption. Also, in this case, dolomites are characterized by better development of both specific surface and porosity (Table 7).

The parameters of the porous texture presented in Table 7 and the surface morphology of the sorbent grains presented in Figure 13a,b and Figure 14a,b show that the thermal dissociation process in the case of both dolomite and industrial sorbent leads to the expansion of the parameters of the porous texture, while both the specific surface and porosity of the dolomite are definitely better developed. The specific surface area of the decarbonated dolomite reaches the value of 18.57 $m^2$/g, and the industrial sorbent the value of 3.14 $m^2$/g. In the case of effective porosity, the differences are visible in particular for the range of pore diameters from 0.01 to 10 μm, considered as sorption pores [27]: dolomite—42.76 $m^2$/g; industrial sorbent—28.16 $m^2$/g.

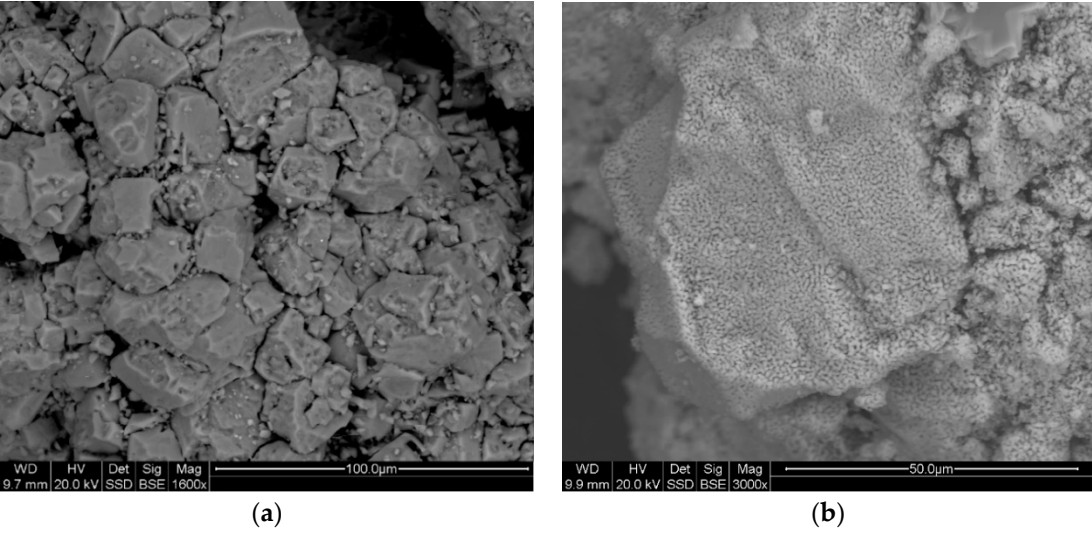

(**a**)　　　　　　　　　　　　　　　　　　　(**b**)

**Figure 13.** The SEM image of the dolomite surface morphology from the Chruszczobród II deposit before decarbonation (**a**) and after decarbonation (**b**), in the light of backscattered electrons.

After the $SO_2$ sorption process, the surface value decreased to 0.21 $m^2$/g—in the case of dolomite and 0.52 $m^2$/g—industrial sorbent (Table 7), suggesting that the dolomite surface was used more effectively (covered of sulfate). The effective porosity of the sulfated samples is high, especially of dolomite—55.39%. The value of this parameter, in combination with the size of the average pore diameter, is very favorable, due to both the decarbonation process ($CO_2$ release from the calcite structure) and $SO_2$ sorption, especially in industrial conditions, when these processes occur almost at the same time. These dependencies are illustrated by photographs showing the surface morphology of the sulfated sorbent grains (Figure 15a,b).

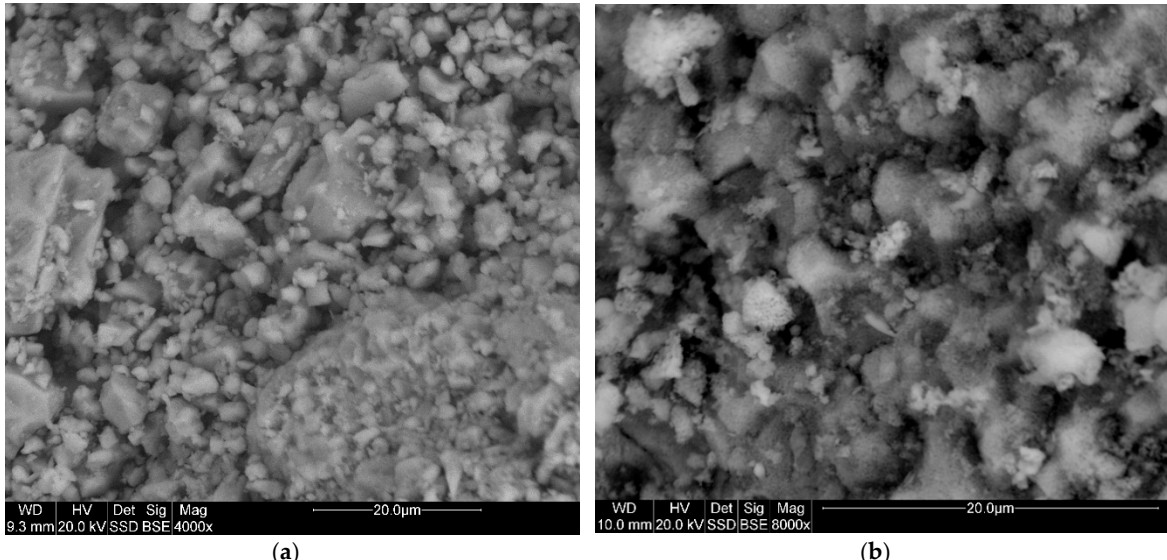

**Figure 14.** The SEM images of the industrial sorbent morphology before decarbonation (**a**) and after decarbonation (**b**), in the light of backscattered electrons.

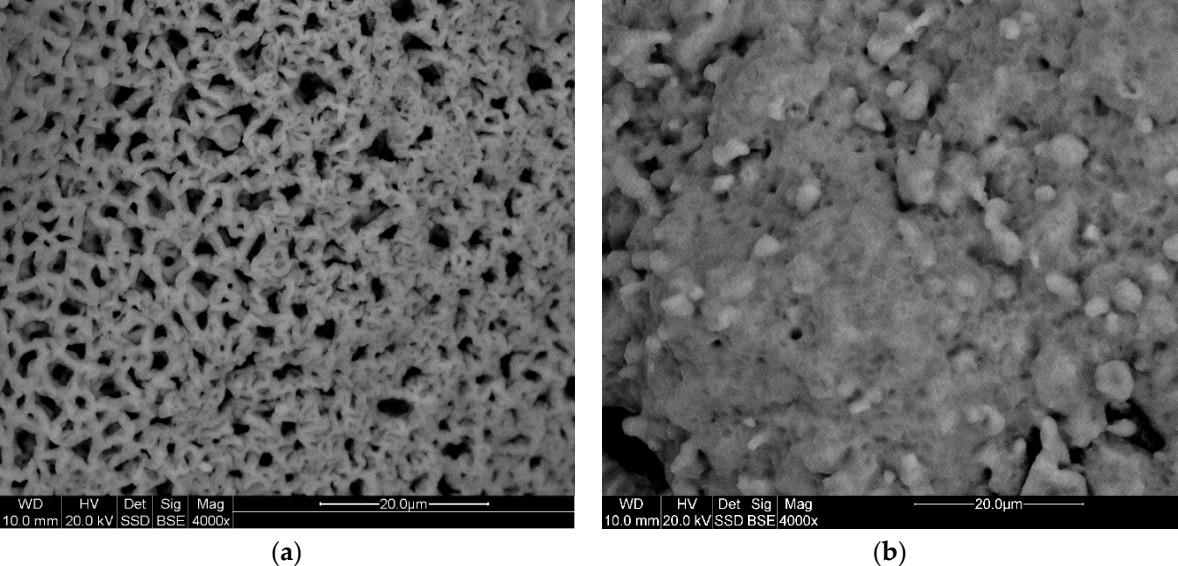

**Figure 15.** The SEM images of the surface morphology of desulfurization products produced on the surface of dolomite grains from the Chruszczobród II deposit (**a**) and industrial sorbent (**b**) in the light of backscattered electrons.

The dolomite grains were covered with sulfate with a clearly porous texture (Figure 15a). The pores are open, and their diameters reach sizes of up to several micrometers. On the industrial sorbent grains' surfaces, sulfate crusts were formed, with fewer open pores and a much smaller diameter (Figure 15b). The porous texture of the calcium sulfates produced on the sorbent grains' surface (Figure 15a) guarantees the free flow of $SO_2$, ensuring a uniform sulfation process of the grains [17]. If a sulfate with a limited porosity is formed on the sorbent grains' surface (Figure 15b), the diffusion of $SO_2$ in the initial desulfurization stage will be stopped, which will result in a low degree of sorbent utilization, manifested by the presence of unreacted particles (Figure 8), and under industrial conditions also with undissociated inner parts of the sorbent grains [30,31].

### 3.4. The Share of Magnesium in the Process of SO₂ Binding and Durability of Desulfurization Products Under High-Temperature Conditions

During analyzing the possibility of using dolomites as $SO_2$ sorbents in the conditions of fluid combustion technology, the most controversial is the thermodynamic stability of desulfurization products containing magnesium under high-temperature conditions [11,13,14]. The share of magnesium in the $SO_2$ binding process is also controversial, and the magnesium oxide (MgO) formed during the thermal dissociation of dolomite is treated as a non-reactive ballast [5,13].

The share of MgO in the $SO_2$ capture process was confirmed by X-ray analysis of desulfurization products obtained as a result of dolomite sulfation. The phase composition of the experimentally sulfated dolomite in the gas-tight retort furnace system presented in the diffractogram clearly indicates the presence of both calcium sulfate–$CaSO_4$ (anhydrite) and calcium and magnesium double sulfate–$CaMg_2(SO_4)_3$ (Figure 16).

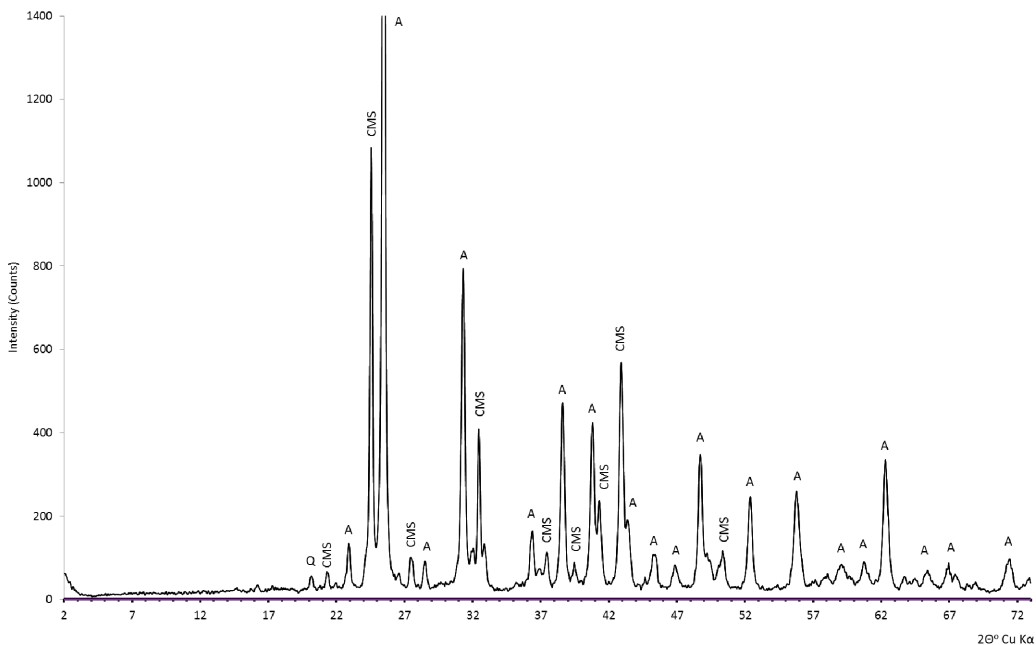

**Figure 16.** The phase composition of the dolomite sulfation product is shown in the diffraction pattern (on the example of dolomites from the Chruszczobród II deposit). Explanations: A—anhydrite ($CaSO_4$), CMS—calcium and magnesium sulfate ($CaMg_2(SO_4)_3$), Q—quartz ($SiO_2$).

The dolomite desulfurization products' durability under high-temperature conditions was tested using thermogravimetric analysis and on the basis of experimentally determined characteristic temperatures of sintering, softening, melting, and flowing. The research results are presented on the example of dolomites from the Chruszczobród II deposit.

The characteristic sintering, softening, melting, and flowing temperatures of dolomite sulfation products indicate their durability under high-temperature conditions, both in oxidizing and reducing atmosphere (Table 8). The spontaneous sintering of the tested material, manifested by the change from loose to weakly bound forms, took place after the temperature exceeded 1100 °C (1150 °C—oxidizing atmosphere and 1130 °C—reducing atmosphere). The softening process, which was accompanied by the transition to a plastic form, was observed under oxidative conditions at the temperature of 1390 °C. The changes up to 1500 °C weren't observed under reducing conditions. Similarly, in the case of melting and flowing points, characteristic changes for this type of process weren't observed in the studied temperature range.

**Table 8.** The characteristic sintering ($T_S$), softening ($T_A$), melting ($T_B$), and flowing ($T_C$) temperatures of dolomite sulfation products presented on the example of dolomites from the Chruszczobród II deposit (°C).

| Characteristic Temperature | Atmosphere | |
|---|---|---|
| | Oxidizing | Reducing |
| $T_S$ | 1150 | 1130 |
| $T_A$ | 1390 | >1500 |
| $T_B$ | >1500 | >1500 |
| $T_C$ | >1500 | >1500 |

The durability of desulfurization products containing magnesium in high-temperature conditions is also indicated by the results of the thermogravimetric analysis. In the temperature range 920–1020 °C with a maximum at 994 °C, a exothermic effect is visible, associated with a weight loss of 11.9% wt. (of which the decomposition of calcite alone accounts for 6.7% wt.—Figure 17). This effect should be associated with the decomposition of calcite, which marked its presence on the diffractograms of dolomite samples after the decarbonation process. The decomposition of $CaMg_2(SO_4)_3$ begins at about 1110 °C (Figure 17). The weight loss associated with the decomposition of this sulfate at 1250 °C is 21.5% wt.

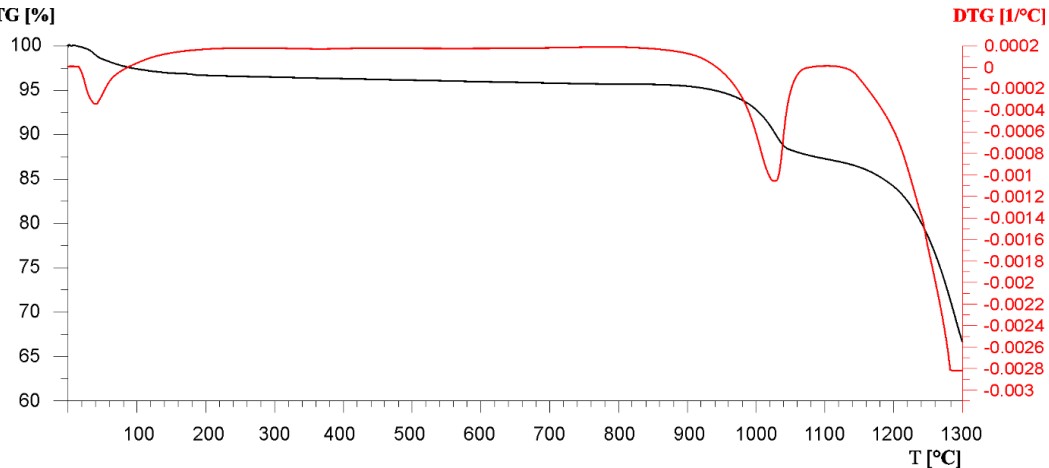

**Figure 17.** The thermal curves show the TG and DTG of dolomite sulfation products from the Chruszczobród II deposit.

## 4. Conclusions

The dolomites can be treated as a potential raw material for the production of $SO_2$ absorbents used in fluidized bed furnaces. The determined values of the reactivity indexes (RI) and absolute sorption (CI) of dolomites are characterized by higher reactivity (RI: 1.8 kmol Ca/kmol S; CI: 174 gS/1 kg sorbent) compared to sorbents produced from limestone (RI: 2.35 kmol Ca/kmol S; CI: 130 gS/1 kg sorbent). The study results showed that the porous textural parameters of the sorbent, produced during the thermal dissociation process, had a decisive influence on the efficiency of $SO_2$ sorption.

The results of the specific surface areas and porous study of samples in their natural state, after the decarbonation and sulfation processes, made by mercury porosimetry, supported by observations using scanning microscopy, gave an excellent image of the texture of the tested dolomites and industrial sorbent, and illustrated the course of the desulfurization process. Dolomites, in comparison to limestones, have more favorable parameters of the porous texture, thanks to which they are able to bind $SO_2$ more effectively. The efficiency of $SO_2$ binding in this case is determined by the size of the

pores, both primary and secondary. The primary pores play the role of $CO_2$ and $SO_2$ diffusion channels, which is important especially in real conditions, where the processes of decarbonation and sorption occur simultaneously. The binding of $SO_2$ takes place on the inner surface of the pores formed during the thermal dissociation of the sorbent, with the production of a layer of calcium sulfates ($CaSO_4$) and calcium and magnesium sulfates ($CaMg_2(SO_4)_3$) with a molar volume greater than the molar volume of carbonates ($CaCO_3$, $CaMg(CO_3)_2$) and oxides ($CaO$, $MgO$). Consequently, this may cause blockage of pores having smaller diameters; as a result, part of the sorbent surface is excluded from the $SO_2$ sorption if the pores have a very small diameter, as seen in the samples of the limestone industrial sorbent. The secondary porosity is determined by the course of the decarbonation process. In the case of dolomites, this process is a two-stage course with the formation of the $CaCO_3$ transitional phase, which is important during the expansion of the porous texture. Another parameter, which affects the $SO_2$ binding efficiency, is the decarbonation temperature. Dolomites compared with limestones are characterized by a significantly lower temperature calcination. Dolomite decomposes completely below the temperature of 800 °C. The calcite decarbonation temperature often exceeds 900 °C, and practical experience shows that to obtain 100% decomposition of $CaCO_3$ into $CaO$ and $CO_2$, the temperature should be kept at 1000 °C. Under the fluidized bed furnaces conditions, it results in lower desulfurization efficiency. The presented research results clearly show that $MgO$ is involved in the binding of $SO_2$ to form a double calcium and magnesium sulfate $CaMg_2(SO_4)_3$. It wasn't possible to precisely determine the Ca: Mg ratio in the resulting sulfate, due to too fine a formation of crystals and a significant share of calcium sulfate in the sulfation products. It was shown, that the desulfurization products containing magnesium in their composition are stable in temperature conditions typical for fluidized bed furnaces. This is indicated both by the sintering, softening, melting, and flowing temperatures (determined under oxidizing and reducing conditions), which are characteristic for the products of dolomite sulfation and the results of thermogravimetric analysis. Thermal transformations of magnesium-containing sulfation products begin after the temperature exceeds 1100 °C.

*The Possibilities of Expanding the Domestic Base of Raw Materials for the Production of $SO_2$ Sorbents Based on Dolomites*

The studied dolomites came from both documented deposits in the industrial category: Chruszczobród II and Rędziny, as well as block and broken stones: Romanowo Górne, Żelatowa. It should be assumed that potentially each documented deposit of dolomite in these categories may constitute a base for the production of $SO_2$ sorbents used in fluid combustion technology. The research results indicate that dolomites with a low calcite content will be predisposed for use as $SO_2$ sorbents. This is indicated by the studies of dolomites from the Żelatowa deposit, which are characterized by a variable content of calcite and can be classified both as dolomites, limestone dolomites, and even dolomitic limestones. In the case of these dolomites, the values of the reactivity index (RI) were characterized by significant differences from 1.82 to 2.45 kmol Ca/kmol S.

It should be assumed that there is also scope to expand the national raw material base for the production of $SO_2$ sorbents based on dolomite waste. The research results on dolomite waste from landfills located in the Rędziny mine (RI: 1.66 kmol Ca/kmol S) i Żelatowa (RI: 2.01 kmol Ca/kmol S), as well as waste dolomites from the Sieroszowice copper mine (RI: 1.58 mol Ca/mol S) indicate that they can be considered as a potential raw material for the production of this type of sorbents [26]. Predisposed for use as this type of sorbents will also be thicker grain fractions coming from the milling of dolomite for the production of dolomite flour, especially since the mill is using industrial dolomites with the highest quality parameters.

Preliminary results of the research on the sorption properties of dolomite waste after the flotation of Zn and Pb and Cu ores, indicate the possibility of their use as $SO_2$ sorbents in fluidized bed furnaces. In this case, Zn–Pb ore flotation waste collected in the Trzebionka sediment pond in Trzebinia was investigated (RI: 2.29 kmol Ca/kmol S), as well as that from the current production of the Copper Ore Enrichment Plant of KGHM Polska Miedź S.A. (RI: 2.12 kmol Ca/kmol S). However, the problem of

metal contamination remains unresolved, which at the moment disqualifies these wastes from the possibility of such use.

**Author Contributions:** Concepcialization, E.H.; methodology, E.H., T.R. and M.S.; validation, E.H., T.R. and M.S.; investigation, E.H., M.S.; resources, E.H. and M.S.; writing—original draft preparation, E.H., T.R and M.S., writing—review and editing, E.H. and M.S.; visualization, E.H and M.S. All authors have read and agreed to the published version of the manuscript.

**Funding:** This research received no external funding.

**Acknowledgments:** This work was carried out in the statutory research of the Department of Mineralogy, Petrography and Geochemistry of AGH in Krakow (No. 16.16.140.315) and the Institute of Mineral and Energy Economy of the Polish Academy of Sciences in Krakow. The research were made with the support of the apparatus of the AGH Center of Energy

**Conflicts of Interest:** The authors declare no conflict of interest.

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
