# Peer review of "Dolomites as SO2 Sorbents in Fluid Combustion Technology"

_resources, doi:10.3390/resources9100121_

Round 1

Reviewer 1 Report

The manuscript is worth of publication. However, authors need to check the terminology consistency in the introduction section related to fluidized bed combustion technology. Some examples are:

“fluid combustion technology” obviously instead of fluidized bed combustion technology, “pressure fluidized deposits” and so.

Author Response

The consistency of the terminology was checked and improved. Introduced “fluid combustion technology” instead of “fluidized bed combustion technology” and “pressure fluidized deposits” and the similar.

Reviewer 2 Report

The authors proposed a study on the possible utilization of dolomite as SO2 sorbent in fluidized bed combustion system. From my point of view, the manuscript has the potential for pubblication but revision fro its improvement is mandatory. My comments and suggestions are reported below:

keywords: le keywords are redundant, the first three are quite similar. I suggest to the authors to change them with other more diversified.

The introduction is quite bare about the literature review. I suggest to expand reporting the works on the utilization of dolomite in fluidized bed for desulphuritazion processes. For this reason, I recommend, for example the following work:

THE INFLUENCE OF SORBENT PROPERTIES AND REACTION TEMPERATURE ON SORBENT ATTRITION, SULFURUPTAKE, AND PARTICLE SULFATION PATTERN DURING FLUIDIZED-BED DESULFURIZATION, by Montagnaro and Salatino, Combust. Sci. andTech.,174(11&12): 151^169, 2002, https://doi.org/10.1080/713712949

Limestone and dolomite as sulfur absorbents under pressurized gasification conditions, by Yrjias at al. Fuel Volume 75, Issue 1, January 1996, Pages 89-95. https://doi.org/10.1016/0016-2361(95)00204-9

Fluidized bed calcium looping cycles for CO2 capture under oxy-firing calcination conditions: Part 2. Assessment of dolomite vs. limestone, by coppola et al. Chemical Engineering Journal Volume 231, September 2013, Pages 544-549. https://doi.org/10.1016/j.cej.2013.07.112

In the materials and methods the descirption of experimental apparatus is absent. It is not reported the device features used for the decarbonization and sulphation of materials. Is it a fluidized bed, a fixed bed or something else? The gas adopted for materials preparation is not reported (nitrogen? Air?) and also the gas used for sulphation test is not completely characterized (SO2, O2 and CO2 and the rest?).

In section 3.2 the statement ‘’ Dolomites can be treated as a potential raw material for the production of SO2 sorbents used in fluidized bed furnaces’’ is quite strong because one of the problem of the dolomite is its high propensity to attrition and fragmentation phenomena as demostrated for redziny dolomite in ‘’Fluidized bed calcium looping cycles for CO2 capture under oxy-firing calcination conditions: Part 2. Assessment of dolomite vs. limestone’’. Could be interesting to study a possible effect of textural properties and attrition and fragmentation phenomena. I suggest to the authora to comment this statement to the light of above considerations.

About the section 4.1, honestly I do not understand its location at the end of the manuscript being a survey of the dolomite distribution in Polan. Probably it is more appropiate to discuss this aspect into the introduction, among other things, this can be exploited to further motivate the study.

Some detected typos:

Page 3, lines 86-87: the word ‘index’ is missing for the definition of RI.

Page 6, line 167: please correct ‘’fot’’ into the brackets, same typo in page 7, line 195.

Page 7, line 196: ‘’ The of pyrite framboidal occurrences’’ probably a word is missing in this sentence

Page 7, line 197: please correct ‘’ crystalswere’’

Page 8, line 217: ‘’allochemhemic components’’ probably should be ‘’allochem components’’

Page 8, line 217: ‘’ zooids and pellets’’ probably should be ‘’ooids and pellets’’

Page 8, line 220: ‘’ calcite and dolomite are covered iron hydroxide compounds’’ please correct the sentence

Author Response

  1. Keywords have been improved as suggested.
  2. The introduction supplemented with the listed references except “Limestone and dolomite as sulfur absorbents under pressurized gasification conditions, by Yrjias at al.”. This article concerns the sorption of H2S and not SO2, and the reaction product is CaS and not calcium or calcium and magnesium sulphates.
  3. The materials and methods contain information on the description of the experimental equipment and the composition of the gas used for the experiment.
  4. The statement “Dolomites can be treated as a potential raw material for the production of SO2 sorbents used in fluidized bed furnaces”.
  5. The attrition and fragmentation were commented on based on the article in Chapter 1. Introduction - lines 66-71
  6. The overview of the distribution of dolomites in Poland has been moved to the introduction as suggested by the Reviewer.

Reviewer 3 Report

Dear authors,

I gave several comments on your work, please look at attached PDF. Also, I found that one previous work is over-transferred into this work, please take a look on the same place (Plagscan result).

The best,

Reviewer

Author Response

  1. The sorption was replaced with absorption, explaining that a new solid phase of sulphate character is formed.
  2. The text includes the figure with the sampling location marked – Figure 1.
  3. “dominant mineral component in relation to both calcite” - is correct. The authors give a phase component - dolomite, composed of CaCO3 and MgCO3.
  4. In fact not, especially secondary dolomites where most pores are re-crystallised.

The matasomatosis process observed in secondary dolomites causes the expansion of porosity, because replacing Ca - Mg reduces the volume.

  1. Information about the analyzed dolomites is included in the introduction.
  2. Plagscan result - The cited work of E. Hycnar is similar of research methodology and the way of presenting research results. This article is about limestones.

Other comments in the text have been corrected.
